

# Influence of alluvial slope on avulsion in river deltas

Octria A. Prasojo[1,2], Trevor B. Hoey[3], Amanda Owen[1] and Richard D. Williams[1]

[1]School of Geographical and Earth Sciences, University of Glasgow, University Avenue, Glasgow, G12 8QQ, United Kingdom
[2]Geoscience Study Program, Faculty of Mathematics and Natural Sciences (FMIPA), Universitas Indonesia, Depok 16424, Indonesia
[3]Department of Civil and Environmental Engineering, Brunel University London, Uxbridge, UB8 3PH, United Kingdom

*Correspondence to*: Octria A. Prasojo (octria.prasojo@glasgow.ac.uk)

**Abstract.** Changed hydrological regimes, sea-level rise, and accelerated subsidence are all putting river deltas at risk across the globe. One mechanism by which deltas may respond to these stressors is that of avulsion. Decades of delta avulsion studies have resulted in conflicting hypotheses that avulsion frequency and location are primarily controlled by upstream (water and sediment discharge) or downstream (backwater and sea-level rise) drivers. Here we use Delft3D morphodynamic simulations to test the upstream-influence hypothesis by varying the initial alluvial slopes upstream of a self-formed delta plain within a range ($1.13 \times 10^{-4}$ to $3.04 \times 10^{-3}$) that is representative of global deltas and recording avulsion, while leaving all other parameters constant. Avulsion timing and location were recorded in six scenarios modelled over a 400-year period. We measured independent morphometric variables including avulsion length, delta lobe width, bankfull depth, channel width at avulsion, delta topset slope and sediment load and compare these to natural and laboratory deltas. We find that larger deltas take more time to avulse as avulsion timing scales with avulsion length, delta lobe width and bankfull depth. More importantly, we also find a strong ($p<0.05$) negative correlation between delta topset slope and avulsion timescale. We argue that topset slope is directly dependent on the varying upstream alluvial slope which determines sediment supply to the delta. Increases in upstream alluvial slope raise transport capacity so bringing more sediment into a delta plain, leading to higher aggradation rates and, consequently, more frequent avulsions. These results induce further debate over the role of downstream controls on delta avulsion.

## 1 Introduction

River deltas are home to ~339 million people worldwide, are hotspots for biodiversity, and crucial carbon sinks (Ericson et al., 2006; Hackney et al., 2020; Loucks, 2019; Shields et al., 2017; Syvitski and Saito, 2007). However, the geomorphic dynamism of river deltas has been altered by amplifying stressors such as change in hydrologic regimes, sea-level rise, and accelerated subsidence, putting human and other systems that rely on river deltas at considerable risk (Giosan et al., 2014; Syvitski et al., 2009; Tessler et al., 2015; Wallace et al., 2014). A frequently observed and geologically rapid mechanism by which deltas respond to these stressors is by flow avulsing from one distributary channel into another. Delta avulsion location has been shown to correlate with backwater length, slope break and valley exit location measured from the shoreline (Ganti et al., 2016a;



Hartley et al., 2017; Prasojo et al., 2022). Many studies have also proposed different hypothesis over the main controls of delta avulsion frequency (e.g. Aslan et al., 2005; Brooke et al., 2020; Edmonds et al., 2009; Kleinhans & Hardy, 2013; Nijhuis et al., 2015; Slingerland & Smith, 2004). However, there is currently no consensus over the conditions under which the various driving factors control delta avulsion frequency.

During avulsion, flow is abruptly diverted out of an established river channel into a new course on the adjacent floodplain or delta plain (Jones and Schumm, 2009; Slingerland and Smith, 2004). When a delta channel avulses, the population and economic activities on the delta plain can be put at risk. Avulsions may be considered rare, but this is partly due to anthropogenic controls on many delta channels preventing avulsion (e.g. built riverbank), and unmodified systems can exhibit avulsion over decadal or sub-decadal timescales, for example once every 12 years in the Yellow River Delta (Jerolmack, 2009)

or 4 years in Sulengguole River, China (Li et al., 2022). Avulsions may be full, where the flow following a new course completely abandons its parent channel, or partial in which only a portion of the flow is diverted (McEwan et al., 2023). Avulsion can be effectively instantaneous but may also be gradual as in the Rhine-Meuse delta that took 1250 years to complete (Stouthamer & Berendsen, 2001). There are also several styles of avulsion: annexation, in which a pre-existing channel is reoccupied; incision, where a new channel is scoured into the floodplain surface as a direct result of the avulsion; and

progradation, where extensive sediment deposition, such as a mouth bar, causes flow bifurcation and formation of a multi-channelled distributive network (Slingerland and Smith, 2004).

River deltas are initiated through repeated mouth bar deposition due to sudden expansion and deceleration of a sediment-laden jet of water entering relatively still water, usually a sea or lake (Bates, 1953; Edmonds et al., 2011; Kleinhans et al., 2013; Wright, 1977). Mouth bars grow in both upstream and downstream directions from the point of initiation. Once a mouth bars

aggradation reaches 40-60% of the initial flow depth, it will stop growing because the sediment flux is advected around the mouth bar rather than accelerated over the bar (Edmonds and Slingerland, 2007; Fagherazzi et al., 2015; Kleinhans et al., 2013). This is the point where avulsion by progradation (or bifurcation) starts in a river delta. Simultaneously, avulsion by incision takes place in the proximal parts of a delta plain when mouth-bar deposition and stagnation induce parent channel backfilling or in-channel aggradation, triggering an avulsion to create a smaller distributive channel network by breaching the

channel levee (Ganti et al., 2016a). The location of the point furthest upstream where a delta channel starts to avulse has been shown to correlate with the location of a break in bed slope (Prasojo et al., 2022; Ratliff et al., 2021), the limit of the backwater zone (Brooke et al., 2022; Ganti et al., 2016a), and the exit point from the river valley (Hartley et al., 2017).

A strong correlation has been identified between the locations of breaks in delta slope and avulsion nodes from 105 global river deltas (Prasojo et al., 2022). Consequently, it is hypothesised that the slope of the alluvial river upstream of a delta

controls the frequency of avulsion on delta plains, with steeper alluvial slopes leading to more frequent avulsions. This control is due to greater sediment transport capacity on steeper slopes (Bagnold, 1966) such that, subject to sediment availability, more sediment per unit width will be delivered to a delta plain where alluvial slopes are steeper. Assuming constant channel width and in the absence of subsidence, any reduction in stream power across the delta plain leads to aggradation, the rate of which will be greater when upstream sediment supply is higher, which in turn leads to increased avulsion frequency (Jerolmack &





Mohrig, 2007; Mohrig et al., 2000). Alternatively, lower alluvial slopes are associated with lower sediment input flux and hence less frequent avulsion.

To test if the upstream of a delta controls the frequency of avulsion on delta plain, we use Delft3D morphodynamic simulation software to: 1) assess the effect of varying alluvial slopes upstream of a delta slope break on the avulsion timescale; and, 2) investigate the primary controls over delta avulsion. Morphometric variables (delta lobe width, channel width at avulsion,

avulsion length, topset slope, bankfull depth and sediment supply) were measured at every timestep during delta growth. These morphometric properties are measured as independent variables expected to covary with avulsion timescales. This investigation aims to: (1) identify the role of alluvial slope upstream of delta plains on avulsion timescales; (2) explain the mechanisms by which the controlling variables determine avulsion timescale; and, (3) compare avulsion timescales from this numerical model with an analytical solution and also with observations from natural and physical experimental river deltas. A

robust understanding of these processes has practical implications due to their direct impact on coastal and inland flood risk on highly populated river deltas, as well as contributing to a fundamental understanding of natural delta building processes.

## 2 Methods

We designed a set of numerical experiments to model a natural scale river delta (7.5 x 7.5 km, 300 by 300 computational cells, each 625m$^2$) using Delft3D (v.4.04.02) software. For comparability with previous studies, we adopted physical parameters

used in similar Delft3D river delta models by Edmonds & Slingerland (2010) and Caldwell & Edmonds (2014). Model bathymetry was designed to accommodate the six alluvial slopes defined below as our model scenarios.

### 2.1 Scenario definition

The model uses a range of alluvial slopes upstream of the delta's slope break ($S_{alluvial}$) (Fig. 1a), which are considered to be representative of natural river deltas (Fig. 1b). Representative percentiles of the ratio between $S_{alluvial}$ and downstream delta

topset slope ($S_{topset}$) were determined from 105 global river deltas measured by Prasojo et al. (2022) (Fig. 1b; Table 1). Percentiles of the $S_{alluvial}/S_{topset}$ ratio of 2.5, 10, 25, 50, 71 and 75 were used to define model scenarios. Model alluvial slopes were calculated from these ratios using a constant downstream slope ($S_{topset}$ = 0.000375) similar to that of the Atchafalaya Bay, Mississippi delta, Louisiana (Edmonds and Slingerland, 2010). During the simulation, both alluvial ($S_{alluvial}$) and downstream topset ($S_{topset}$) slopes evolve downstream from the inlet, as aggradation occurs even though equilibrium has been reached at the

model inlet (Video S1). We define equilibrium to have been reached when there is constant sediment discharge and channel depth at the model inlet.



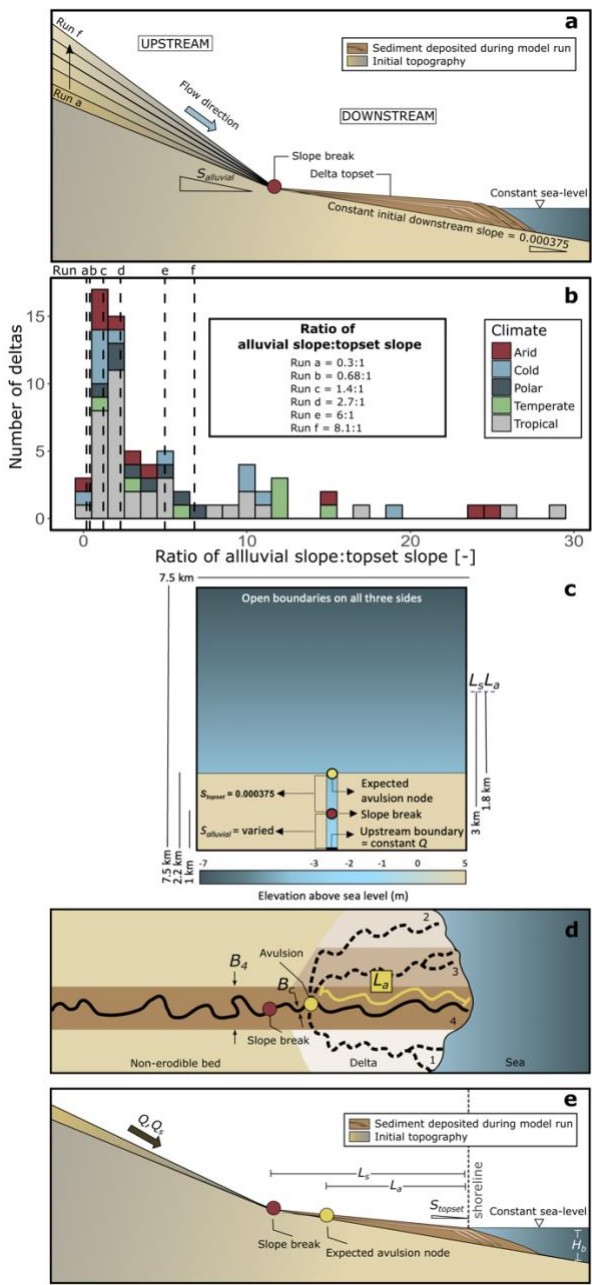

**Figure 1:** (a) Schematic diagram of the model design. The alluvial slope of each run was calculated from six percentiles from the alluvial slope-topset slope ratios of modern river deltas shown in Fig. 1b. Initial downstream slope is kept constant at 0.000375, adopted from the downstream slope of the modern Mississippi delta (Edmonds and Slingerland, 2010). (b) Distribution of the ratio between alluvial ($S_{alluvial}$) and topset slopes ($S_{topset}$) from 105 modern river deltas distributed across five climate regions. Ratios used for numerical model runs are indicated by vertical dashed lines. (c) Plan view of the model design. $L_s$ and $L_a$ are slope break and avulsion lengths, respectively. The non-erodible bed at 5 m above sea level represents non-erodible bedrock. (d) Schematic diagram of a river delta showing avulsion location, inlet sediment supply ($Q_s$), lobe width of each avulsion ($B$), avulsion length ($L_a$) and channel widths measured at avulsion ($B_c$), modified from Chadwick et al. (2020). Numbers near the shoreline represent the number of delta lobes that were used to measure $B$; e.g. $B_4$ on (d) represents





the width of the fourth lobe built. (e) Schematic cross-section showing basin depth ($H_b$) and topset slope ($S_{topset}$). Parameters shown in Fig. 1d-e are measured at each timestep during delta growth.

## 2.2 Model setup

We use Delft3D software to model six scenarios. Delft3D is a physics-based model that simulates hydrodynamics and
morphodynamics (Edmonds & Slingerland, 2010; Caldwell & Edmonds, 2014; Nienhuis et al., 2018a;b) and has been validated
for a wide range of environments, including self-formed river deltas (Edmonds & Slingerland, 2007, 2008; Geleynse et al.,
2011; Morgan et al., 2020; Nijhuis et al., 2015; Rossi et al., 2016; Williams et al., 2016). Flow is computed using depth-
averaged, nonlinear, shallow-water equations obtained from three-dimensional Reynolds-averaged Navier-Stokes equations
(Edmonds and Slingerland, 2010). The modelled velocity distribution is then used to compute sediment transport and to update
the bed elevation according to divergence in sediment transport (Caldwell and Edmonds, 2014).

**Table 1:** Numerical modelling scenarios as defined in Fig. 1.

| Run ID | Percentile from $S_{alluvial}$ to $S_{topset}$ ratio | Initial alluvial slope, $S_{alluvial}$ | Initial downstream topset slope, $S_{topset}$ | Ratio of alluvial slope to downstream topset slope |
|--------|-----------|----------------|----------------|----------------|
| a | 2.5 | $1.13 \times 10^{-4}$ | $3.75 \times 10^{-4}$ | 0.30 |
| b | 10 | $2.55 \times 10^{-4}$ | $3.75 \times 10^{-4}$ | 0.68 |
| c | 25 | $5.25 \times 10^{-4}$ | $3.75 \times 10^{-4}$ | 1.4 |
| d | 50 | $1.01 \times 10^{-3}$ | $3.75 \times 10^{-4}$ | 2.7 |
| e | 71 | $2.25 \times 10^{-3}$ | $3.75 \times 10^{-4}$ | 6.0 |
| f | 75 | $3.04 \times 10^{-3}$ | $3.75 \times 10^{-4}$ | 8.1 |

We adopted physical parameters from a previous synthetic self-formed river delta numerical model ('scenario o') from
Edmonds & Slingerland (2010) and Caldwell & Edmonds (2014) (Fig. 1c). The model is rectangular with four boundaries, the
incoming river discharge being located at the 'South' boundary of the model and the other three boundaries set to 0 m elevation
above sea level (Fig. 1c). The constant incoming river discharge, set at 1050 m³ s⁻¹, is uniformly distributed across the 250 m
wide inlet channel, and inlet sediment discharge is in equilibrium with transport capacity. Various alluvial slopes are achieved
by having various inlet channel elevation in each run while maintaining the receiving basin's depth. Consequently, sediment
discharge varies in each run because of the varied alluvial slope as the main controlled variable in the experiments. Our
modelled deltas closely represent natural deltas because the discharge ratio and the differences in bed heights between
bifurcating distributary channels follow ranges similar to those reported for natural deltas (Edmonds & Slingerland, 2010).
Sea-level remains constant within the model, and no subsidence, tide or wave effects are considered.

The model domain is 7.5 km x 7.5 km to avoid the delta plain extending across the model boundaries. We introduce a slope
break 1 km from the inlet boundary to drive delta formation in the model's initial bathymetry. Using the slope break-avulsion
length scaling identified from global river deltas (Prasojo et al., 2022) the expected first avulsion node location should emerge



in each scenario at around 2.2 km from the inlet (Fig.1c). Constant sediment grain-size distributions are used throughout the model ($D_{50}$ = 125 µm with a Gaussian distribution) introduced as non-cohesive sediment and medium-grain silt ($D_{50}$ = 30 µm) introduced as cohesive sediment, the critical bed shear stress for erosion = 0.10 N m$^{-2}$, and the model initially has 5 meters of fully mixed sediments. Other physical and numerical parameters were held constant across all scenarios (Table 2).

For 18 days simulation, the model produces one output every 480 minutes. Hence at the end of simulation, the model stores 52 visualisation outputs (i.e. maps). Using a morphological scale factor (*morfac*) of 175, these 52 maps represent 3150 days (8.6 years) of prototype time. Because bankfull discharge occurs for c.2% of time on average (Dunne & Leopold, 1978), 18 days of simulation thus represents around 430 years of 'real' time (i.e. 8.6 years divided by 0.02).

**Table 2:** User-defined model parameters (adopted from Edmonds & Slingerland (2010); Caldwell & Edmonds (2014)).

| Parameter | Value | Units |
|---|---|---|
| Grid size | 300 x 300 | cells |
| | 7.5 x 7.5 | km |
| Cell size | 25 x 25 | m |
| Run duration | 18 | days |
| 'Real time' converted run duration | 430 | years |
| Basin bed slope (downstream of slope break) | 0.000375 | (-) |
| Initial channel dimension (width x depth) | 250 x 2.5 | m |
| Upstream non-erodible bed elevation | 5 | m |
| Initial channel length upstream of slope break | 1000 | m |
| Initial avulsion length from the expected shoreline | 1800 | m |
| Water discharge | 1050 | m$^3$.s$^{-1}$ |
| Constant water surface elevation at downstream open boundary | 0 | m |
| Initial sediment layer thickness at bed | 5 | m |
| Number of subsurface stratigraphy bed layers | 1 | (-) |
| Computational time step | 0.2 | min |
| Output interval | 480 | min |
| Morphological scale factor | 175 | (-) |
| Spin-up interval | 1440 | min |


## 2.3. Surface morphological metrics

The model reaches equilibrium after ~3-6 days of simulation time when we begin measurements of morphometric variables and avulsion timing. The time at which avulsion occurs across the delta plain were recorded throughout all timesteps after



reaching equilibrium. Avulsions were defined as any time when a distributary channel produced during delta formation
changed its course and commenced deposition of a new delta lobe. We only consider avulsions caused by progradation or
incision that are common in river deltas (Slingerland and Smith, 2004). The timing of each avulsion in the model was noted
and converted to a 'real' time as $T_{a\ empirical}$.

Numerous morphological surface metrics can be used to describe delta form. The surface metrics used here follow those used
in an analytical solution for avulsion timescale (Eq. (4) from Chadwick et al., 2020), which utilises delta lobe width ($B$),
channel width at avulsion ($B_c$), avulsion length ($L_a$), basin depth ($H_b$), magnitude of relative sea-level rise ($z$), topset slope
($S_{topset}$), bankfull depth ($h_c$) and sediment supply ($Q_s$). Avulsion length, delta lobe width, channel width at avulsion and delta
topset slope were measured on all maps after equilibrium is reached. The delta lobe width, channel width at each avulsion node
and avulsion length were measured in QGIS from the georeferenced images produced by Delft3D (Fig. 1d, Table S1). Delta
lobe width is measured as the maximum width of each lobe, while avulsion length is measured along the longest channel from
the shoreline to the most upstream avulsion node located at the 'expected avulsion node' defined in Fig. 1c. Topset slope
($S_{topset}$) was calculated as the average slope of a delta plain. Digital elevation models (DEM) for each timestep were first
extracted from Delft3D. Afterwards, each DEM was cropped to only cover the delta plain (i.e. offshore area measured 2.2 km
from the model's 'south' boundary). We then filter the DEM to only include elevation, $z$ in between 0-5 meter ($0 < z < 5$) to
cover delta plain that is exposed above the sea-level (i.e. $z = 0$ m) but below the non-erodible bed (i.e. $z = 5$ m; Fig. 1c). DEM
is transformed to be slope raster defined as the change of elevation for each DEM cell. Mean topset slope for each timestep
for each scenario is then extracted as the slope values for each scenario are normally distributed (Table S1). Sediment supply
($Q_s$) at the channel inlet was obtained from a Delft3D visualisation software, QUICKPLOT (v2.60.65942).

Bankfull depth ($h_c$) was calculated using Eq. (1) (Parker, 2007).

$$h_c = \left(\frac{C_f Q^2}{g B_c^2 S_{topset}}\right)^{\frac{1}{3}}, \tag{1}$$

where $C_f$ is a bed friction coefficient [-] = 0.002 for large lowland rivers (Parker et al., 2007), $Q$ = bankfull discharge [m³ s⁻¹]
= 1050 m³ s⁻¹, and $g$ = gravitational acceleration [m s⁻²] = 9.81 m s⁻².

The avulsion timescale was calculated between each successive pair of avulsions observed in the model ($T_{a\ empirical}$) and was
correlated with all the measured morphometric variables (e.g. $Q_s$, $L_a$, $B_c$, $B$, $S_{topset}$, $S_{alluvial}$, and $h_c$) from all post-equilibrium
maps. Scatter plots and Pearson correlation coefficients ($r$) were used to assess the linearity of relationships and potential
dependencies between all variables.

## 3. Results

Figure 2 shows the morphology of the deltas in each scenario at the final timestep. Overall, the different alluvial slopes produce
delta plains that exhibit different shoreline configurations, different numbers of active distributary channels and slightly



different delta plain sizes. One delta plain reached the model boundary (Run f, Fig. S1) and this scenario was repeated with a larger domain size and the avulsion timescales and morphological metrics were observed from this larger domain.

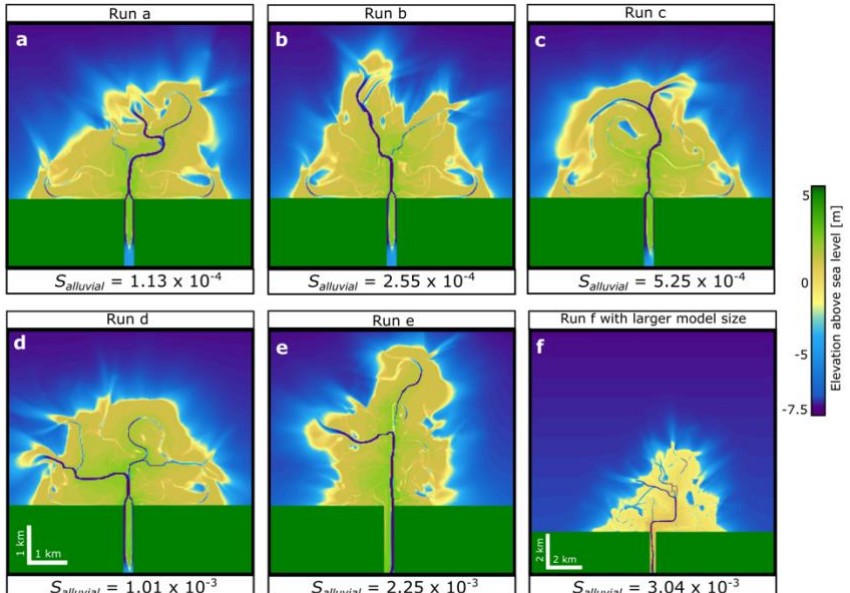


**Figure 2:** (a-f) River deltas for each run at the final simulation timestep. Run f was repeated with a larger (12.5 x 12.5 km) model size to avoid the delta plain reaching the model's boundary. Morphometric measurements for Run f were made on this larger model size.

The first avulsion observed in the model occurs as quickly as 8.27 years (run a & e) with the longest being 298 years (run e). Avulsions occurred 7 times in run c, 15 times in run f and 16 times in run a. A skewed distribution of avulsion timescale was

observed in most scenarios (Fig. S2) with the overall median being 297.7 years and mean 278.9 years. The median avulsion timescale remains unchanged over runs a-d (Kruskal-Wallis, $p > 0.05$), although the variance in this timescale decreases as slope increases (Fig. S2). Runs e and f show significantly more frequent avulsions (shorter times between avulsions) (Kruskal-Wallis, $p < 0.05$). Notably the variance in avulsion period increases noticeably between runs d and e (Fig. S2).

Fig. 3 shows correlations between observed avulsion timescales in the model ($T_{a\ empirical}$) and the independent morphometric

variables measured in each timestep. $T_{a\ empirical}$ has a significant negative correlation with topset slope, $S_{topset}$ ($r=0.86$; N=232), implying that avulsions occur more frequently in steeper delta topset slope. $T_{a\ empirical}$ is also well correlated with bankfull depth, $h_c$ ($r=0.71$) as bankfull depth defines the aggradation thickness necessary for avulsion to occur. The greater the bankfull depth, the more sediment that is needed to fill the channel to induce avulsion, hence increasing the avulsion timescale. Avulsion timescale is also well correlated with the delta size represented as delta lobe width, $B$ ($r=0.63$), delta avulsion length, $L_a$

($r=0.58$) and channel width at avulsion, $B_c$ ($r=-0.58$), suggesting that a larger delta takes more time to avulse. $T_{a\ empirical}$ is only moderately correlated with sediment load ($Q_s$, $r=-0.45$) and initial upstream alluvial slope ($S_{alluvial}$, $r=-0.27$). However, because sediment load ($Q_s$) is mainly controlled by the initial upstream alluvial slope ($S_{alluvial}$) with $r=0.8$, sediment load may be responsible for defining the delta topset slope, $S_{topset}$ ($r=0.41$) that then dictates the avulsion timescale ($T_{a\ empirical}$).



Significant but weaker correlations are also found between other morphometric variables, such as $B_c$-$B$, $B_c$-$L_a$, $B$-$L_a$, $S_{alluvial}$-$L_a$, because as a delta grows, the delta plain and its constituent channels enlarge in an allometric manner, as observed in natural, physical laboratory and numerical deltas (Wolinsky et al., 2010). Additionally, initial upstream alluvial slope ($S_{alluvial}$) covaries with the avulsion length, $L_a$ ($r$=0.65), suggesting that higher transport capacity in steeper alluvial slope may produce a longer delta, consistent with the findings from global river deltas (Prasojo et al., 2022). Bankfull depth ($h_c$) also has high correlation with channel width at avulsion, $B_c$ ($r$=-0.95) and topset slope, $S_{topset}$ ($r$=-0.7) as defined from Eq. (1).

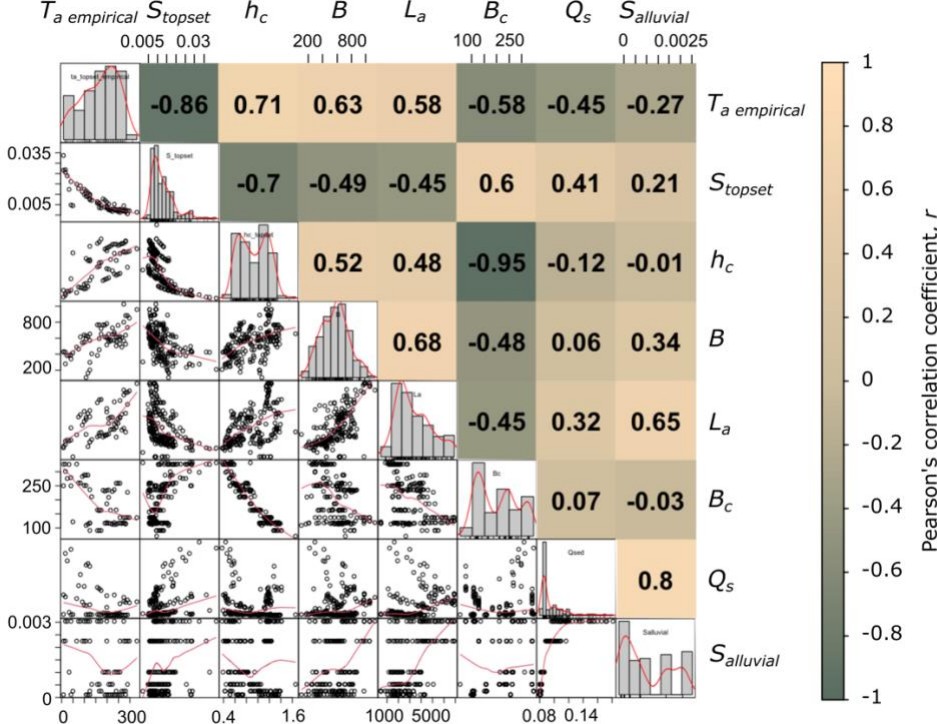

**Figure 3:** Pearson correlations (upper right panels) between avulsion timestep ($T_{a\ empirical}$) and independent morphometric variables along with their distributions (diagonal panels) and correlations (lower left panels). Units on this figure are years for $T_{a\ empirical}$, m³ s⁻¹ for $Q_s$, and meters for $B_c$, $h_c$ and $L_a$. $S_{topset}$ and $S_{alluvial}$ are dimensionless. Note that $h_c$ is autocorrelated with $S_{topset}$ and $B_c$ as shown in Eq. (1) and $S_{alluvial}$ is initial alluvial slope available from Table 1 as the independent variable used to define the six experimental scenarios. Red lines on the correlation plots are LOWESS curves. Red lines on the frequency distributions are to aid visualising distributions.

Fig. 4 shows the data from the model and ordinary least square regressions for the most significant correlations found in Fig. 3. The regressions are statistically significant and have relatively narrow confidence bands (grey shaded areas in Fig. 4), although the data exhibit both scatter and some clustering. Avulsion timescale is inversely correlated with topset slope ($R^2$=0.77, $p$<0.05) (Fig. 4a) and positively correlated with bankfull depth ($R^2$=0.55, $p$<0.05) (Fig. 4b). Avulsion timescales increase with the size of the delta, represented by delta lobe width (Fig. 4c), avulsion length (Fig. 4d) and channel width at avulsion (Fig. 4e). Lastly, avulsion timescale is weakly correlated ($R^2$=0.16, $p$<0.05) with sediment load measured at the model inlet (Fig. 4f).



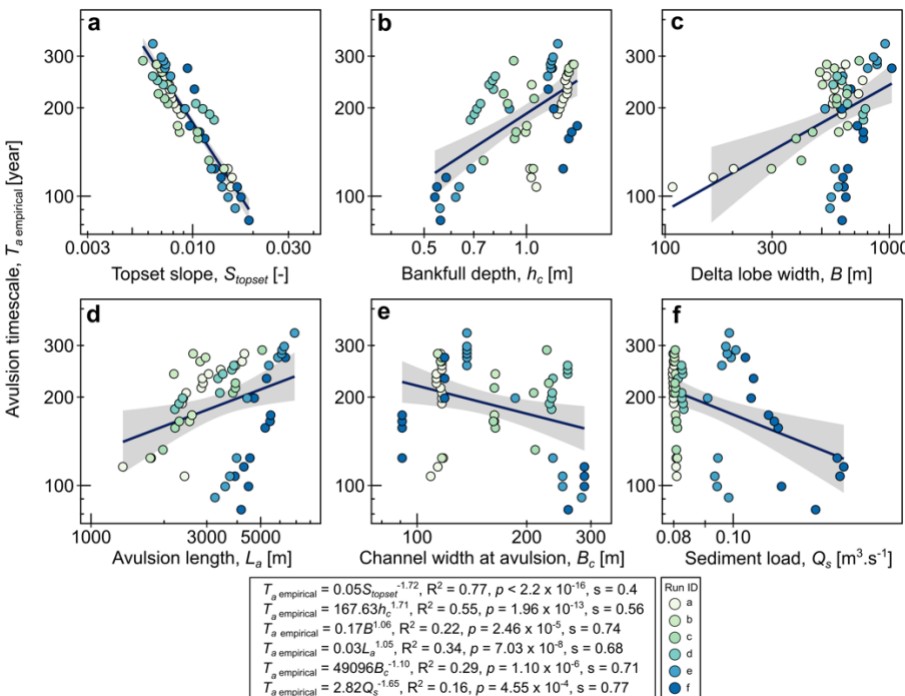

**Figure 4:** Regressions of independent morphometric variables against avulsion timescale ($T_{a\ empirical}$) observed in model runs with 95% confidence band in grey (N = 233). Note that data exhibit scatter and some clustering (c,f) that may indicate alternative patterns of morphological adjustment. S = standard error of the residuals.

Fig. 5 shows the detail of how one avulsion occurs in a delta plain to illustrate the process. An avulsion is initiated by an increase of in-channel deposition in a distributary channel around the proto avulsion node (Fig. 5a-b). In-channel aggradation increases the likelihood of overbank flows as it elevates the distributary channel above the surrounding floodplain. Aggradation continues until the levee is breached as the surrounding delta plain provides an easier path for a distributary channel to flow (Fig. 5c). This newly avulsed channel then distributes more water and sediment away from the initial distributary channel path (Fig. 5e-g). A significant increase in post-avulsion sediment thickness around the avulsion node characterises how sediment is now distributed to this newly avulsed channel (Fig. 5h). Note also that bifurcation can occur at the same time as an avulsion develops. Bifurcation is initiated by a mouth bar deposition in a more distal part of the delta plain (Fig. 5b). The distributary channel feeding this mouth bar bifurcates once the depth of mouth bar is ~40% of the initial basin depth, consistent with findings from Edmonds & Slingerland (2007). However, we are uncertain if the mouth bar deposition forces an upstream wave of in-channel deposition that leads to channel being unstable to trigger an avulsion as the sediment thickness along the avulsing distributary channel remains similar through time (Fig. 5a-g).





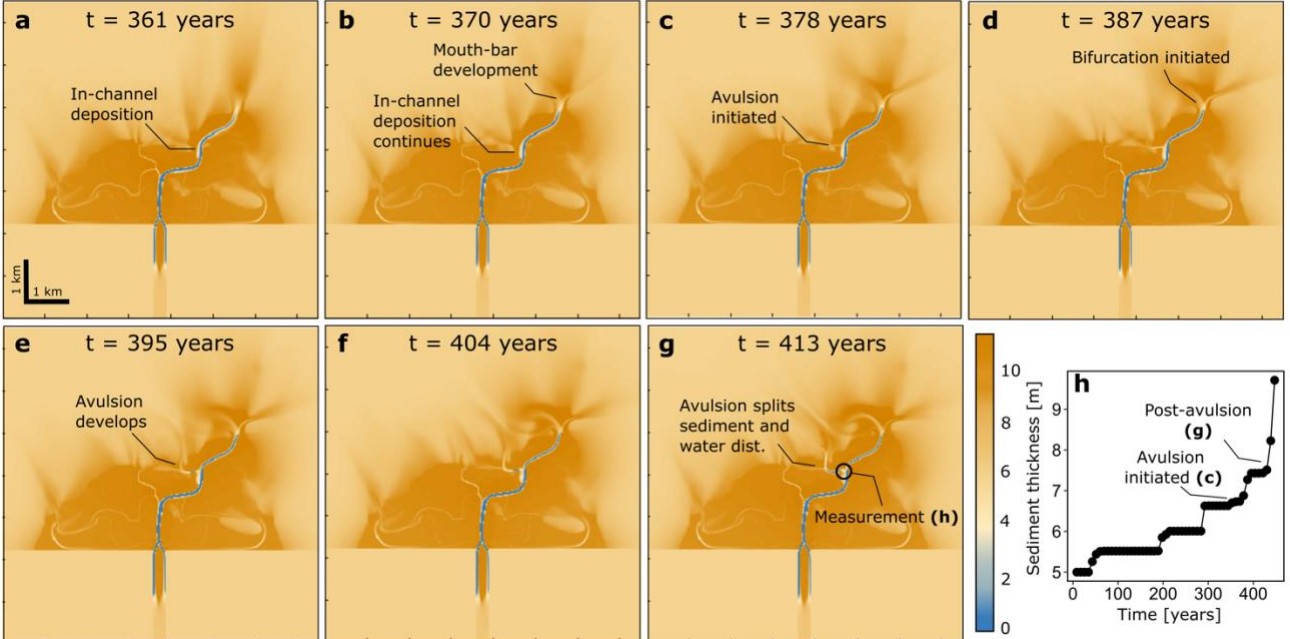

**Figure 5:** Time series images from Run a showing the detail of an avulsion process (a-g). Time series sediment thickness at the avulsion node (h).

## 4. Discussion

Since avulsion is infrequent, it is difficult to acquire data sets on avulsion frequency from field studies. The six scenarios modelled in this study have merit in providing a large data set from which to infer the process controls over avulsion timescales.

### 4.1. Investigating variables controlling the avulsion timescales

In this study, only the initial alluvial slope ($S_{alluvial}$) was varied. All the other measured variables are from these experiments, where deltas were allowed to self-form. Our experimental set-up determines that initial alluvial upstream slope controls how much sediment enters the delta plain ($r$=0.8, Fig. 3, Table 3). A steeper initial alluvial slope ($S_{alluvial}$) has a higher transport capacity and so transports a greater sediment load ($Q_s$), that is then deposited on the delta plain. Higher sediment load produces a higher vertical aggradation rate ($v_a$) in the distributary channel (Chadwick et al., 2020) which elevates the distributary channel floor above its surrounding floodplain. The higher the distributary channel floor relative to the floodplain, the easier it is for an avulsion to occur (Jerolmack and Mohrig, 2007; Mohrig et al., 2000). In summary, we propose that the avulsion timescale in our model is influenced by the amount of sediment deposited in a delta plain. While sediment load is controlled by the steepness of the initial alluvial slope transporting that sediment. Other investigations that have also found that sediment mass-balance is the primary control on avulsion timescales include a radially averaged model (Muto, 2001; Muto & Steel, 1997), a channel-averaged model (Reitz et al., 2010), and backwater-scaled models (Chadwick et al., 2019; Moodie et al., 2019).



**Table 3:** Median avulsion timescale along with average sediment load and initial alluvial slope for each scenario. The median avulsion timescale is used in preference to the mean as the distribution of avulsion timescale is skewed (Fig. 3).

| Run ID | Initial alluvial slope, $S_{alluvial}$ [-] | Average sediment load, $Q_s$ [m³.s⁻¹] | Median avulsion timescale, $T_{a\,median}$ [year] |
|--------|--------------------------------------------|---------------------------------------|---------------------------------------------------|
| a | $1.13 \times 10^{-4}$ | 0.0804 | 335.0 |
| b | $2.55 \times 10^{-4}$ | 0.0809 | 322.5 |
| c | $5.25 \times 10^{-4}$ | 0.0814 | 339.1 |
| d | $1.01 \times 10^{-3}$ | 0.0835 | 330.8 |
| e | $2.25 \times 10^{-3}$ | 0.0973 | 252.2 |
| f | $3.04 \times 10^{-3}$ | 0.1243 | 181.9 |

On the other hand, sediment load is also responsible for shaping the profile of a delta topset slope, which is why avulsion timescale ($T_{a\,empirical}$) is also correlated with the delta topset slope ($S_{topset}$, $r=0.86$, Fig. 3). A strong correlation between avulsion timescale ($T_{a\,empirical}$) and delta topset slope ($S_{topset}$) resembles avulsion behaviour in comparable alluvial sedimentary environments (e.g. alluvial fans and fan deltas). Diverse fan experiments have shown that avulsion timescale is influenced by fan-channel gradient (equivalent to delta topset slope in our model) (Schumm et al., 1987; Whipple et al., 1998; Van Dijk et

al., 2012; Leenman & Eaton, 2021). Fan-channel slope is dependent on sediment flux (Parker et al., 1998; Bagnold, 1986) and sediment flux also influences the avulsion timescale as shown earlier in our model (Table 3). Consequently, avulsion timescale also has a strong correlation with the fan-channel slope or delta topset slope. However, we propose that delta topset slope is a causal effect of the amount of sediment fed into a delta plain in our model. As the initial alluvial slope ($S_{alluvial}$) controls the sediment load ($Q_s$) feeding a delta plain which in turn determines $S_{topset}$, we argue that $S_{alluvial}$ plays more fundamental role than

$S_{topset}$ in influencing the avulsion timescale ($T_{a\,empirical}$) observed in our model.

## 4.2. Comparison with analytical solution and natural deltas

Chadwick et al's (2020) mass-balance based analytical solution is used to calculate expected avulsion timescales for our model conditions (Eqs. (3-6), Table S1). Measured independent morphometric variables are used in Eqs. (3-6) to calculate avulsion frequency ($f_a$) and timescale ($T_a$).

$$f_a = \frac{1}{T_a} = \frac{1}{(1-\lambda_p)} \frac{Q_s}{(L_a - D)BH + DB\left(H_b + z + \frac{DS_{topset}}{2}\right)} \; if \; D \geq 0, \tag{3}$$

$$f_a = \frac{1}{T_a} = \frac{1}{(1-\lambda_p)} \frac{Q_s}{L_a BH} \; if \; D < 0, \tag{4}$$

$$D = (H - z)/S_{topset}, \tag{5}$$

$$H = H^* h_c, \tag{6}$$



with $f_a$ = avulsion frequency [year$^{-1}$], $Q_s$ = sediment load [m$^3$ s$^{-1}$], $\lambda_p$ = sediment porosity [-], $L_a$ = avulsion length [m], $D$ = delta lobe-progradation distance [km], $B$ = delta lobe width of each avulsion [m], $H$ = aggradation thickness necessary for avulsion [m], $H_b$ = basin depth [m], $z$ = magnitude of sea level rise [m], $S_{topset}$ = topset slope [-], $H^*$ = avulsion threshold [-], and $h_c$ = bankfull depth [m] calculated using Eq. (1).

In calculating these analytical avulsion timescales, sensitivity analyses were undertaken using avulsion thresholds ($H^*$) of 0.2,
0.5, and 1.4, which are realistic for lowland deltas (Ganti et al., 2019), and $D > 0$ since there is no allogenic forcing that would make the delta regress. The analytical avulsion timescales for $H^*$ = 0.2, 0.5, and 1.4 are $T_{a\,H^*=0.2}$, $T_{a\,H^*=0.5}$, and $T_{a\,H^*=1.4}$, respectively (Table S1). Since sea-level is constant in this study, sea level rise $z = 0$. Sediment porosity ($\lambda_p$) is assumed to be 0.4 (Jerolmack, 2009; Paola et al., 2011), bed friction coefficient ($C_f$) = 0.002 for lowland rivers (Parker et al., 2007), and constant bankfull discharge ($Q$) = 1050 m$^3$ s$^{-1}$. Analytical avulsion timescales were then compared to avulsion timescales
observed from 19 natural river deltas, two fan deltas and one downscaled physical laboratory fan delta documented in Chadwick et al. (2020) and Jerolmack & Mohrig (2007), using topset slope values from Prasojo et al. (2022) (Table S2).

Figure 6 shows avulsion timescale scaling relationships observed from our model compared to natural deltas, a physical laboratory fan delta and analytical solutions. Generally, the pattern and magnitude observed from our numerical model are in good agreement with both the analytical solution and natural deltas. The negative correlation between avulsion timescale and
topset slope found in our numerical results is similar to natural and physical laboratory deltas, even though our topset slope is steeper than in most natural deltas (Fig. 6a). Positive correlations between bankfull depth, avulsion length, delta lobe width and avulsion timescale are also consistent between our model and the natural and physical laboratory deltas (Fig. 6b-d). However, channel width at avulsion, sediment load and avulsion timescale do not show clear patterns (Fig. 6e,f).

We consider that results from our analytical-numerical model and natural-physical laboratory deltas are directly comparable,
but care is needed in their interpretation. The analytical calculations are for fixed values of input variables, and field data are snapshots assumed to represent equilbria. Conditions change during our numerical model runs, and topset slopes reduce through time (Fig. S3a) which would be expected to lead to an increase in avulsion timescale as the delta grows. This results from gentler topset slopes having reduced transport capacities, so reducing the in-channel aggradation rate as explained above (Fig. 6a). Similarly, as bankfull depth is also calculated based on the topset slope value (Eq. (1)), disagreement between analytical and numerical model results is expected (Fig. 6b). Moreover, delta lobe width ($B$) and avulsion length ($L_a$) in the
original analytical solution are assumed to be constant with $B=40B_c$ and $L_a=0.5L_b-0.2L_b$ (Chadwick et al., 2020). As we found that delta lobe width ($B$) and avulsion length ($L_a$) grow through time (Fig. S3b,c) in our numerical model, analytical model assumptions lead to this disagreement (Fig. 6c,d). Our numerical model shows that avulsion takes longer in a larger delta (Fig. 4c, d and Fig. 6c,d).



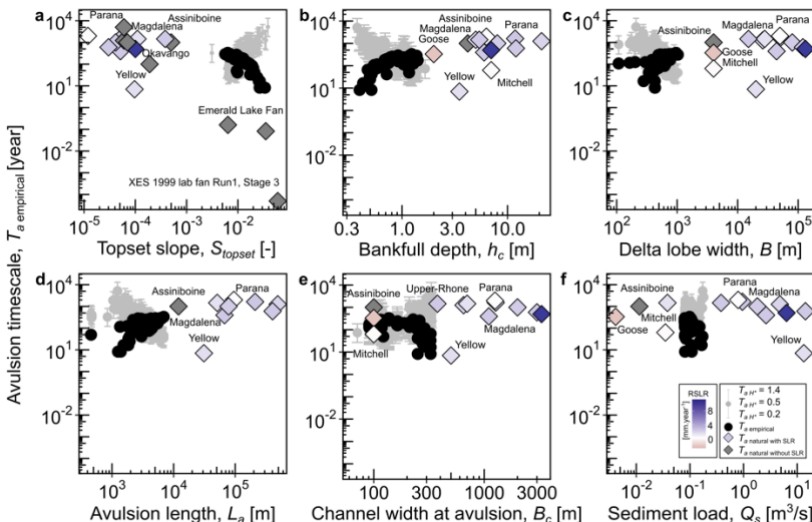

**Figure 6:** Relationships between avulsion timescales and independent variables: (a) topset slope; (b) total sediment load; (c) channel width at the avulsion location; (d) delta lobe width; and (e) bankfull channel depth from model, analytical equations and natural deltas. The plots show model values ($T_{a\ empirical}$) and those calculated from analytical equations (Eqs. (3-6)). Solid black circles are empirical results from the model. Grey dots and bars are results from the analytical equations using three avulsion threshold $H^*$ values ($T_{a\ H^* = 1.4}$, $T_{a\ H^* = 0.5}$, $T_{a\ H^* = 0.2}$). Diamonds are results from the natural and laboratory deltas: grey diamonds have no information about their relative sea-level changes; purple diamonds are for deltas with relative sea-level rise ($RSLR$; mm yr$^{-1}$) colour-coded as shown. Data from natural deltas and the laboratory experiment are available in Table S2.

The negative correlation between $Q_s$ and $T_a$ from our numerical model (Fig. 4f) deviates from empirical data gathered from natural deltas (Fig. 6f). Since natural deltas have relatively larger delta plains than our numerical models as shown from avulsion length, bankfull depth and delta lobe width values (Fig. 6b-d), we normalised sediment discharge to delta size by dividing $Q_s$ to ($L_a.B/2$), which is a vertical aggradation rate, $\eta$ [m s$^{-1}$] . Vertical aggradation rate is then normalised again by dividing it to shear velocity, $u^* = \sqrt{gh_cS_{topset}}$ [m s$^{-1}$] as we are looking at a vertical aggradation rate related to transport capacity of flow to remove sediment from a delta plain with no subsidence, $\eta^*$ [-]. Fig. 7 shows a better negative correlation between $T_a$ and $\eta^*$ from both natural deltas and our models, suppporting our argument on sediment discharge influence on avulsion timescale. The range of initial alluvial slopes used in the model, although covering the slopes from 105 global deltas, are too small to produce the entire range of $Q_s$ inputs to natural deltas (Fig. 6f). Fig. S4 shows that a significant increase in $Q_s$ (and a resultant decrease in avulsion timescale) would require the initial alluvial slope to be 6 times the initial downstream slope (Table 1, 3). Further simulations, using a different initial downstream slope value or more varied slope ratios, could enable a larger range of $Q_s$ to be covered and hypothetically enable a closer agreement between natural deltas and numerical models (Fig. 6f).



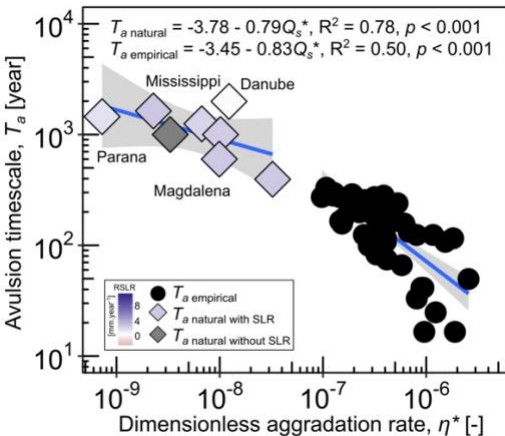

**Figure 7:** Relationship between avulsion timescale ($T_a$) and dimensionless aggradation rate ($\eta^*$) observed from our natural deltas and our models. Regression lines are shown in blue with 95% confidence band in grey.

The bathymetry of the basin is defined at the beginning of our model runs (Fig. 1). The bathymetry adjusts during a model run to reach an equilibrium profile at a simulation timestep > 3-6 days (Video S1). In comparison to other models in which "reference profiles" and their evolution are defined by making assumptions (e.g. floodplain deposition is assumed to be equal to riverbed aggradation) (Chadwick et al., 2019; Moodie et al., 2019; Moran et al., 2017; Ratliff et al., 2018; Edmonds et al., 2022), our approach directly evaluates avulsion frequency and location that emerge from the physics of self-forming delta-lobes. Avulsions in our models consistently arise from channel superelevation (Fig. 1c), consistent with our previous global empirical study (Prasojo et al., 2022).

However, we should also acknowledge that the ratios between alluvial and topset slope gathered from natural deltas (Table 1) are assumed to be in dynamic equilibrium with the environmental conditions in our study. Even though most modern global deltas have developed since the early Holocene, inevitable natural and anthropogenic changes of boundary conditions such as changes in sediment or water discharge and local SLR must have happened during this period (Stanley & Warne, 1994). On the other hand, the slope data derived from remote sensing only represents a single snapshot of this property. Consequently, our measured topset slopes from natural deltas may be a transient response to this changing boundary condition. Repeating the work with different initial topset slope presents an opportunity for further investigation.

The avulsion timescales calculated for natural deltas ($T_{a\ natural}$) do not correlate well with relative sea-level rise rate ($RSLR$) (Fig. 6a-f and Fig. S5) (Chadwick et al., 2020). This result supports the hypothesis that the frequency of avulsions may be unaffected by sea-level rise, as also found in an earlier numerical model study (Ratliff et al., 2021) and a global empirical study (Colombera & Mountney, 2023). We propose that avulsion frequency and location are dominated by upstream forcing (i.e. alluvial slope or catchment sediment supply, Fig. 5) (Prasojo et al., 2022) rather than downstream forcing by sea-level rise or backwater effects (Fig. S6) (Chadwick et al., 2020; Chatanantavet et al., 2012; Ganti et al., 2016b). As our deltas are self-formed and evolve throughout the simulations, avulsion and backwater lengths (as a function of topset slope, Eq. (1)) grow linearly and each scenario has a unique avulsion-backwater length ratio (Fig. S6), rather than being constant as has been



suggested (Ganti et al., 2016; Chatanantavet et al., 2012). This consequently challenges our current understanding as to the circumstances under which *RSLR* and backwater length may control avulsion frequency in river deltas. Deltas with low sediment supply, shallow basin, or varied discharge have shown that *RSLR* and backwater length strongly define their avulsion timing (Chadwick et al., 2019; Ganti et al., 2016).

Previous literature on the relationship between the frequency of avulsion and sea-level rise is somewhat equivocal. A field study conducted on the Mitchell River delta, Australia found that avulsion frequency increases with sea-level fall (Lane et al., 2017). Numerical model results suggest that avulsions on the Mississippi (faster) and Trinity (slower) Rivers showed different responses to Holocene sea-level rise even though they are geographically adjacent (Chatanantavet et al., 2012; Moran et al., 2017). An example during sea-level fall from the Goose River delta, Canada, shows that avulsion frequency remained constant

during this base-level adjustment (Nijhuis et al., 2015). In contrast, avulsion frequency in the Rhine-Meuse delta, Netherlands, increased during the Holocene sea-level rise period (Törnqvist, 1994), possibly due to aggradation rate ($v_a$) being controlled by *RSLR*. Our experiments do not address this issue, and we propose that further investigations combining numerical and flume experiments that are based on observations from natural deltas may aid resolution of this debate.

## 4.3. Implications for delta management

Our modelling results advance our understanding of how sediment input from the catchment regulates the frequency of avulsions in river deltas. The complex hydraulic and sediment transport processes that lead to the correlation between alluvial slope and avulsion timescale are linked to sediment load, the rate of in-channel aggradation and how hence rapidly channels become perched. Consequently, with the increase of anthropogenic forcing both directly within river deltas and throughout upstream catchment areas (Best, 2019; Darby et al., 2015; Dunn et al., 2019; Hackney et al., 2020), river delta managers can

use sediment load management to reduce the risk of avulsion-driven flooding. Interventions that control sediment load may be more effective than those which address other factors that are less closely correlated with avulsion frequency, such as flood variability, delta size, or channel morphology (Aslan et al., 2005; Brooke et al., 2020; Edmonds et al., 2009; Nienhuis et al., 2018; Slingerland & Smith, 2004; Valenza et al., 2020).

However, finding a perfect balance between reducing avulsion frequency, maintaining the sediment load required to nourish

delta environments and to hinder deltas' risk from subsidence and coastal erosion, is challenging. In some locations, deforestation that increases sediment supply is responsible for c.25% of net land gain on global deltas, which also hastens future avulsions (Nienhuis et al., 2020). Conversely, river impoundment is responsible for more than a 50% reduction in sediment delivery to the global ocean since 1950, collectively leading to a loss of a delta land of $127 \pm 8.3$ km$^2$ annually over the past 30 years (Nienhuis et al., 2020). This declining sediment input not only poses threats to the long-term sustainability

of deltas but also renders them susceptible to adverse effects from rising sea levels and ecological degradation due to sediment starvation and saltwater ingress (Jordan et al., 2019). Therefore, gaining insights into the distribution patterns and quantities of sediments in deltas is imperative to ensure their continued sustainability.



### 4.4. Next steps

An important extension of this modelling work is to have more varied $S_{topset}$:$S_{alluvial}$ ratios, water discharge ($Q$) and sediment
load ($Q_s$), as variability in these may affect the geomorphic processes controlling avulsion timescale. Multi-temporal
observation of well-studied natural river deltas, such as the Yellow (Moodie et al., 2019), Mississippi (Chamberlain et al.,
2018) or Rhine-Meuse (Pierik et al., 2018) deltas, could then be used to validate model results. Moreover, incorporating other
variables such as grain size and sediment cohesion, forcing through sea-level rise and subsidence, varying basin geometry and
adding vegetation that controls crevassing and consequently increases avulsion timescale in future numerical modelling should
be considered (Nienhuis et al., 2018; Pierik et al., 2023; Sanks et al., 2022). In particular, considering the importance of
projected global sea-level changes and the variability of results reported in the literature, a better understanding of sea-level
rise impacts on delta avulsion is needed.

We have used a simplified modelling approach and have isolated one predictor variable while holding other factors constant.
Observations of the processes and evolution in the numerical deltas shows the complexity of hydraulic and morphodynamic
processes across delta plains. Future work will need to address this complexity, including: (a) How does the forcing studied
here (alluvial slope and consequent sediment input) interact with a combination of other factors (e.g. sea-level, wave and tidal
regimes, and anthropogenic effects)? (b) How do the other controls (e.g. $Q_s$, $Q$, riverbank material, vegetation) in river deltas
influence avulsion timescales? and, (c) how might these avulsion signals be preserved or shredded in the rock record?

### 5. Conclusion

We conducted a suite of numerical morphodynamic modelling experiments with variable river alluvial slopes (from $1.13 \times 10^{-4}$ to $3.04 \times 10^{-3}$) to understand the controls over avulsion location and timescale in a river delta. Sediment load, controlled by
alluvial slope upstream of a delta plain, shapes the delta topset slope that shows the highest correlation with avulsion timescale.
We propose that alluvial slope upstream of a delta plain is the dominant variable influencing the avulsion timescale.
Mechanistically, this is due to greater sediment transport capacity over steeper alluvial slopes leading to increased sediment
input to the delta plain, accelerated vertical aggradation and more frequent avulsion. The results support the hypothesis of
upstream forcing influencing delta avulsion timescale and location, rather than downstream influence by backwater length or
sea-level rise. However, our model has several limitations such as dynamic equilibrium assumption from our alluvial-topset
slope ratios, homogeneous initial topset slope adopted from Mississippi delta and constant discharge applied in the model. A
robust understanding of the main factors controlling avulsion in river deltas has significant implications due to their direct
impacts on: (i) coastal and inland hazards on highly populated river deltas; and, (ii) rock record interpretations.

**Data availability**

The morphometric variables and avulsion timescales observed from our models are available in Table S1. The dataset from natural and laboratory river deltas used in this study (Table S2) and model scenarios (Run a-f) along with their simulation videos of bed level change and non-cohesive and cohesive sediment concentration distributions from Run a are available in
the FigShare repository (Prasojo et al., 2023a,b, 2024).

**Author contributions**

OAP, TBH, AO and RDW conceptualised the study. OAP and TBH designed the Delft3D simulation. OAP collected the morphometrics from numerical model and natural deltas. OAP wrote the paper, and TBH, AO and RDW reviewed it. All authors discussed the results and contributed to the final article.

**Competing interests**

The authors declare that they have no conflict of interest.

**Acknowledgements**

We thank A.J.F. (Ton) Hoitink, Luca Colombera, Andrew Moodie and other referees of an earlier version who have contributed significantly to improve the quality of this manuscript. For the purpose of open access, the authors have applied a Creative
Commons Attribution (CC-BY) licence to any Author Accepted Manuscript version arising from this submission.

**Financial support**

This study was funded by an Indonesia Endowment Fund for Education (LPDP) awarded to Prasojo.

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
