# Peer review of "Influence of alluvial slope on avulsion in river deltas"

_EGUsphere, 2024_

## Referee Comment (RC1)

[referee-annotated manuscript omitted]

---

## Referee Comment (RC2)

[referee-annotated manuscript omitted]

---

## Author Comment (AC2)

Dr Octria Adi Prasojo
School of Geographical & Earth Sciences
University of Glasgow
octria.prasojo@glasgow.ac.uk

Dear Stephan Toby,

We would like to thank you for all the meticulous comments, questions and detailed corrections you gave on our manuscript 'Influence of alluvial slope on avulsion in river deltas'. Following reviewers' comments, we have adjusted how we present some of our data which will allow us to clarify the interpretation of the figures. Please see below the following responses to your comments and questions. Line numbers below are based on the original pdf file that you commented upon. We cluster your similar questions into one so that it is easier to follow. We are hopeful that our responses and edits will clear up elements of the manuscript that were confusing. In the text below your comments are in black font and our responses are in blue font.

1. L16: backwater length sets an avulsion node in which larger delta means larger backwater length and lower slope means larger backwater length.
L21: This sentence doesn't justify all the nice results mentioned before. There's a nice summary of upstream influences, but no disprove of downstream controls. So end with a statement that's close to what this manuscript demonstrates.
L336: This study does not challenge those controls.
This study has a nice systematic analysis of relations between avulsion frequency vs slope control, which is a proxy for sediment supply. It does not challenge other mechanisms.
L391: again, I haven't seen evidence here that disproves a backwater control.

Thank you for your comment about the size of a delta in relation to the backwater length. We agree that larger delta with a lower slope will have a longer backwater length. In our work, we find that in larger deltas, avulsions will happen slower as there is more accommodation space in a distributary channel to be filled with sediment, before an avulsion can happen (Fig. 6b-d). We also agree that in a larger delta, the backwater length also grows larger (Fig. S5). However, this finding contradicts what was found by Ganti et al. in his 2016 Science paper. In that paper, they found that the ratio of avulsion length and backwater length remain constant (i.e. $L_a$:$L_b$ = 0.5) after a certain duration of simulation during a variable discharge experiment (their Fig. 5). In contrast, in our experiments (left figure below) we find that the avulsion length:backwater length ratio never reaches a constant value of 0.5 . We also vary the discharge under equilibrium and non-equilibrium states in our *in-prep* manuscript, using an intermittency factor from the Minnesota river ($I = 0.12$). Similarly, we vary the alluvial slope to be gentle, medium and steep. With varying discharge, we consistently find that backwater lengths grow as the avulsion timescale gets longer (right figure below). This is the reason why we mention in our abstract that our results induce further debate over the role of downstream controls (e.g. backwater effect, SLR) on delta avulsion.

[Figure]

2. L37: This paragraph and terminology need to be cleared up. Are we talking about delta lobe scale avulsions, bifurcations, or river avulsions at the scale of a reach, not in a delta-setting. Although both are called avulsions, different features are meant different timescales may be involved.

Thank you for requesting this clarification. We have revised the main text to clarify what we meant. In this study, we focus on delta lobe-scale avulsions and specifically avulsion by progradation and incision.

3. L59: I don't yet see the 'consequential' relation here. To me the break in slope could be due to the exit point from the river valley or related to the backwater zone.
L65: Not quite, there should be some relative term in here, this doesn't work in absolute terms as other dimensions of the system matter too. E.g. the mississippi delta receives (received) lots of sediment influx in absolute numbers, but has a low gradient.

We apologise for the confusion made in our sentences. We have added extra words in that paragraph to improve the focus and clarity. By assuming that the width of the alluvial river upstream of a delta has a constant value (L62), we could expect that more sediment will be delivered to a delta plain in a steeper alluvial slope, leading to a more frequent avulsion on delta plain. Higher sediment load delivered to a delta plain also causes a steeper delta topset slope, as we observe in our model (Fig. 6a). We do not intend to explain the cause of the slope break in this paper. We suggest to the reviewer to refer to our previous paper, where we discuss in more detail about the processes contributing to slope break-avulsion length scaling that we find to be consistent from global deltas: https://doi.org/10.1029/2021GL093656.

4. L73: Just to be clear, what's the thing that really matters: upstream slope or upstream sediment supply?

Thank you for raising this important question. We would say that both of them matter. Our hypothesis in this study is based upon our previous finding on consistent scaling between slope break length and avulsion length, measured from delta shoreline analysis from our global dataset. Because of this consistent scaling among 105 deltas across 5 climate regions, we hypothesise that alluvial slope upstream of a delta plain will control the amount of sediment supply fed into the delta plain. This is, of course, under the assumption that the width and depth of the alluvial inlet are constant, with only the slope varied (as we explain in our Methods

section). From varying alluvial slopes, it will induce more sediment being delivered to a delta plain in a steeper alluvial slope. With more sediment delivered to a delta plain, the in-channel aggradation rate will be higher, making an avulsion easier to occur.

5. L90: that's a local equilibrium at the inlet (and not sure where precisely), which is close to the boundary condition and not the most reliable location to use for analysis

Thank you for your comment on the equilibrium state we define in our model. As shown in Table 2, we include a one day spin-up interval and found that constant sediment discharge and channel depth can be reached after 3-6 days of simulation at the model inlet. It means that our equilibrium state is reached after 4-7 days of simulation. Physically, equilibrium state (or normal flow) represents a perfect balance between the downstream gravitational impelling force and resistive bed force, as defined by Parker (2004). A key parameter in describing this equilibrium state can be derived from the Froude number, Fr. As shown from the distribution of Froude number from scenario a (as an example) available here: https://doi.org/10.6084/m9.figshare.27004381.v1 , Froude number in our model is always below 1 (or subcritical flow) everywhere in the delta plain. Hence, we believe it is safe to use constant sediment discharge and channel depth at inlet in defining equilibrium state in our model.

6. Figure 1: why is the slope break at a different point than the avulsion node? Is the avulsion node there because flow can spread into the basin while it's restricted at the slope break?
   I don't see Salluvial in here.

Thank for raising this very important question. We use the scaling between slope break ($L_s$) and avulsion length ($L_a$) measured from global delta shoreline of $L_a = 0.6L_s$ (Prasojo et al., 2022) and adopt this scaling into our numerical model. As you may observe on the right-hand side of Fig. 1c, if we design the $L_a$ to be 1.8 km, then the $L_s$ will be 3 km from the reference point. We have explained the reason of putting the slope break and expected avulsion node in L124-126 and we apologise for the confusion caused.
$S_{alluvial}$ is mentioned in Fig. 1a.

7. L105: Refer to a model description here rather than studies that make use of the model or move all those references to the end of the sentence. Delft3D has tons of uses in many other domains than the delta studies mentioned here.
   L109: This sentence, and others, are almost a copy from what they said, which is bad practice. It's also not an original reference for how the model works, but for an application of the model. If you'd like to reference how the model works, include one on the model rather than the application.
   As I understand the manuscript, the model setup and all parameters are the same as Edmonds and Slingerland? That'd be good to mention here, because for example for sediment transport different equations can be used within D3D.

We appreciate your comment on the way we cite other works. We apologise for the bad practice and have properly cited some original works that explain how the model works. We thank you for your suggestion to state that the model setup and parameters are the same as Edmonds and Slingerland (2010). We have accommodated this in the main text.

8. L147: From experience, it can be quite difficult to determine what's really an active channel, what is an avulsion, where does the lobe stop. What criteria were used?

Thank for your question. Yes, we agree with you that this is difficult. We extracted all bed levels produced for each timestep. From there, we visually searched for a full avulsion that happens on the delta plain. When we found one, we then measured the channel width at the avulsion node and measured the river length from the most upstream avulsion node to the delta shoreline (L150). Defining delta lobe width is subjective because a clear distinction of which lobe is active is difficult to spot. The variable delta lobe width comes from the analytical solution by Chadwick et al., 2020 published in PNAS (https://doi.org/10.1073/pnas.1912351117). They used a constant value of delta lobe width in their analytical model, as we also explain in L286. Using this measurement to analyse numerical modelling results involves a judgment of an active lobe that can be observed through a distributary channel and is active at the moment of measurement. Determining whether a channel is active can be determined by investigating the in-channel bed shear stress values and assessing the presence of a newly deposited mouth-bar offshore.

9. L149: Does that match the real avulsion location?

Not really, although this method follows previous works by Chatanantavet et al., 2012; Ganti et al., 2016; Chadwick et al., 2020 and others. However, as avulsions are clustered around the main active distributary channel (shown from the bed shear stress maps below), measured avulsion lengths will not be far off from the distance between real avulsion location to shoreline.

10. L153: the break in slope between topset and foresets is often below 0m in d3d runs.

Yes, we agree and that is what we observe from our Delft3D model as well. However, we would not mean to include the foreset slope and topset slope below sea-level because the 'autobreak' point or the break in slope between topset and foreset is not always the same as shoreline position at any given simulation timestep. To reduce bias in defining this break, we only include slope measurement from the topset/delta plain deposited above the sea-level.

11. L155: bit lost here. Change of elevation in time or space? If in space, in what direction?

Thank for raising this question. We meant change of elevation in $x$ and $y$ directions. We have added this into the main text.

12. L156: distribution not shown in Table S1.

We apologise for this. We have added Fig. S1 to show the distribution of the topset slope value.

13. L158: why calculated, not measured?

We calculated, instead of measured, the bankfull depth to be consistent with other studies (e.g. Paola, 2000; Ganti et al.,2016; Chatanantavet et al., 2012) and our previous study (Prasojo et al., 2022) that used bankfull depth to approximate the backwater length.

14. L173: Is there a temporal trend in here as the delta grows throughout a run. Are the avulsion with vastly different timescales the same type of avulsions?

We clumped avulsion by progradation and by incision together because of the difficulty in differentiating them in the numerical model. Consequently, we could not identify the temporal trend in avulsion type in our model. However, we provide timelapse videos of bed level change throughout the simulation (Video S1). Early avulsions occur around the expected avulsion node shown in Fig. 1c. However, in later time avulsions progress downstream along the main active distributary channel throughout the delta plain building process. As a delta plain grows larger, the avulsion length also grows longer as shown in Fig. S4c.

15. L206: this is surprising as the abstract mentions that higher alluvial slope means higher supply means higher aggradation means shorter. So I'd expect a strong correlation here?

Strong correlation between avulsion timescale and sediment load was indeed what we expected at the beginning of the simulations. However, as we explain in the Discussions and Fig. 7a, the correlation gets stronger once the sediment load is normalised by the delta size and shear velocity. We did the normalisation on the basis of different delta plain sizes (as shown in Fig. 2) and shear velocity to remove sediment from each delta plain, assuming no subsidence occurs in the delta plain.

16. L214: This is not so clear from the figure, looks more like incision.
    L235: check this, because I don't see it in figure 5.
    As an alternative explanation to everything being a depositional process, can a steeper delta topset lead to higher shear stresses on the levee and floodplain and thereby facilitate more frequent incision and avulsion?

We thank you for your comment. We also observe how incision initiates an avulsion process. However, if we look in detail from Fig. 5a, we could observe that in-channel deposition (i.e. light-yellow colour) occurs just next to the active channel (i.e. blue colour). The in-channel deposition gets more intensive in Fig. 5b before it incises the floodplain as shown in Fig. 5c. Fig. 5h shows time-series sediment thickness at the avulsion point in which it increases significantly once an avulsion occurs at that observation point.
Your alternative explanation is very interesting. We further investigate this alternative explanation by extracting bed shear stress maps shown below. As you may observe, when the delta plain is being built, high bed shear stress occurs at the river bends most of the time. However, if the erosional nature of delta avulsion is true, it is safe to expect that avulsion will happen at most bends where the highest bed shear stress values are concentrated. However, avulsion does not happen at the highest bed shear stress value. At the avulsion point we discuss in Figure 5, the bed shear stress remains the highest at the bend right before the avulsion point. However, avulsion (i.e. floodplain is now incised) then happens not at the highest bed shear stress (i.e. at river bend), but rather at the straight part of the active distributary channel. Once avulsion occurs, there is a zone of low bed shear stress values in the newly avulsed channel. This is due to in-channel deposition that makes the river jump, creating a newly avulsed shallow channel as bed shear stress is a function of river depth. This observation supports the depositional nature of delta avulsion in our study, instead of erosional nature of delta avulsion.

[Figure]

17. L250: While this paragraph draws analogues with fan experiments, it doesn't explain a mechanism for Stopset control on Ta.

    L252: isn't that what the Parker papers etc already said, as was mentioned a few lines above?

We appreciate your input above. As mentioned in L252-253, we propose that delta topset slope is a product of sediment fed into a delta plain that happens to have high correlation with avulsion timescale. This does not imply that steeper delta topset slope causes a more frequent avulsion. In terms of the causalities, we propose that alluvial slope defines sediment load fed into a delta plain that later control the avulsion timescale. The by-product of higher sediment load is a steeper delta plain slope as we try to express using diagram below:

[Figure]

18. L268: Why not a porosity from D3D?

Thank you for this. We initially would like to extract the porosity value from Delft3D. However, as we could not find the way to extract bed porosity value from our model, we maintain the porosity value assumption used in the original analytical solution paper by Chadwick et al., 2020. The 40% bed porosity is reasonable estimate for natural deltas as also found by Parker et al., 2008; Coleman et al., 1998 and Jerolmack, 2009.

19. L273: The figure seems to show opposite trends between the analytical solution and the model runs, most clearly in fig 6a.

Yes, we also observe similar thing. We tried to explain the reason for this in L284-289. Moreover, we put L279-280 to caution that care is needed in interpreting this comparison.

20. L288: If delta size is so important, wouldn't it be fair to vary La and other dimensions with time? So after each avulsion in the model, predict with the analytical equations how long it would take until the next avulsion, and calculate a misfit with the observed avulsion in the model.
Currently the claim is that delta size is important, but the equations omit that so it's not a fair comparison.

We apologise for the confusion. In the original analytical solution made by Chadwick et al. (2020), they (not us) used constant avulsion length as they want to accommodate backwater-scaled avulsion length in their model. However, in our work, we vary avulsion length, as you suggested here, when we calculated the analytical avulsion timescale as can be found from Table S1.

21. L306: Transport theory predicts that bed shear stress and transport are directly dependent on slope, so this is a bit of an odd statement. I guess transport in the model is noisy and obscures the relation especially when measured over short time windows, creating the noise and uncertainty. Not sure if this is a relevant model result with wider implications.

We thank you for your comment. As we mention in L119, alluvial slope is the only controlled variable in our experiment, while all other variables are from self-emergent behaviour. That means that at timestep 0, each scenario in our model has a unique alluvial slope value (Table 1). With the set of varied alluvial slope, Delft3D will then calculate the flow velocity, sediment transport and update the bed levels at each computational timestep. At each timestep, bed mass as the result from suspended and bedload transport divergence is calculated for each cell. The updated bed levels (i.e. derived to be slope) will then be used to calculate flow velocity and sediment transport for the next timestep. Please refer to Delft3D manual Chapter 8.6 for further explanation (https://content.oss.deltares.nl/delft3dfm2d3d/D-Morphology_User_Manual.pdf).

22. L335: Fig S6 does not show a function of topset slope. Also, S6 has a log-log scale, so these are not linear relations.

We apologise for the confusion. What we meant in that sentence is that backwater length = hc/S. While hc is a function of topset slope as shown from Eq. 1. As topset slope gets gentler through time, hc gets larger. With hc gets larger, backwater length gets longer through time. We also thank you for correcting our scale. We have replaced the figure to have linear scale instead of log-log scale.

23. L347: This paragraph on RSLR is redundant, it doesn't add to the storyline of this paper and only raises questions to the reader.

Thank you for your comment. However, we believe that our paragraph about RSLR is relevant to our study. As we explained our hypothesis about downstream control (e.g. RSLR, backwater) of delta avulsion in the Abstract and Introductions, our results show that we do not see the influence of RSLR in making avulsion timescale shorter (i.e. more frequent). Fig. 6 and S5 show that there is barely a correlation to be observed between RSLR values from natural deltas to avulsion timescale. If RSLR makes delta avulsion happens more frequently, Fig. S5

should show negative linear correlation. However, we do not observe this in our study. Hence, we believe that our paragraph about RSLR is still relevant to our study.

    24. L356: This manuscript does not show data or references to support that.

Thank you for correcting this. We apologise for our mistake and have removed that sentence.

    25. L369: agree with the other reviewer's comment here. Also, would Stopset be a consequence of Salluvial? The two seem linked, as its the sediment and water supply in the upstream that ultimately determine downstream.

Yes, we agree. What we meant here is initial downstream slope, not topset slope as shown in Fig. 1a. We have revised this in the main text. We propose that instead of using the initial downstream slope = 0.000375 as adopted from the Mississippi delta, the more variation we have in this initial downstream slope value will be our next step.

    26. L392: this should be in methods or discussion, not conclusion

Thank you for your comment. However, we are afraid that we cannot agree with your comment as we believe it is only fair if we acknowledge the limitation of our study in this section. Hence, we hope you are okay with us keeping this sentence here.

We would like to thank you again for the time and effort you spent.

Best wishes,
Octria Adi Prasojo

17 November 2024

---

## Referee Report (RR1)

**Reviewer reply to the revised manuscript 'Influence of alluvial slope on avulsion in river deltas'**

This is my 2nd review round for this manuscript by Prasojo et al. Previously I mentioned that:

(1) Not all arguments and choices in the design and analysis of this study are clear to me.

(2) Some arguments in the text seem to contradict, and not all claims are clearly supported by the results.

In their revised manuscript, Prasojo et al. improved on the clarity of their model setup and analysis. I would like to thank the authors for their revisions and clarifications in their comments to an earlier review of this manuscript. On a few minor points, I am still not convinced by the evidence presented in the manuscript, mainly concerning claims on downstream controls, the avulsion illustrated in figure 5, and the clarity of the model setup, as detailed below. Overall, the data, analysis and discussion presented in this manuscript is a valuable read to those interested in avulsion controls.

The points below refer to the author's reply to the previous round of review.

**Point 1 and point 23, concerning the debate of downstream controls on delta avulsion.**

The manuscript does not systematically test downstream controls on delta avulsion, yet at several points claims that there is no correlation. In my opinion this is not evidenced in the paper. The authors refer to figure 6 and S5 for this claim. Although those figures contain data on RSLR for various deltas, the manuscript does not test whether avulsion timescales in a single delta will be different under different RSLR scenarios.

**Point 6, Figure 1, and point 9, about the model setup and location of the avulsion node.**

Figure 1c shows that the expected avulsion node is exactly at the end of the valley, bounded by non-erodible bed at 5m above sea level. To me, the manuscript and author's comments do not make clear whether the avulsion node is there because of some scaling between La and Ls, or because of the river is no longer confined by the valley walls. Continuing on the location of avulsion nodes, I don't understand the answer by the authors to point 9 and I'd recommend the authors to clarify throughout the manuscript whether an avulsion length is the real measured length or the length until the expected avulsion node.

**Point 16, about an example of an avulsion in figure 5.**

I very much appreciate the author's efforts to explore an alternative explanation for the avulsion demonstrated in figure 5.

Unfortunately, I still do not see the depositional character of an avulsion in figure 5, other than figure 5h. Figure 5a-g show an incision, as demonstrated by a change from dark orange to blue colours in the new branch. From the colour map, I cannot resolve if and why figure h suggests deposition. From the colour maps, the process seems more incisive rather than depositional.

Finally, a comment about **figure 6**. I noticed data in the figure changed compared to a previous version and it is not clear why.

---

## Author Response (AR3)

Dr Octria Adi Prasojo
School of Geographical and Earth Sciences
University of Glasgow
octria.prasojo@glasgow.ac.uk

Dear Dr Schwanghart,

We would like to thank you for agreeing with Dr Baar on the publication of our manuscript. We also thank you for your detailed final comments on the manuscript. Please see below the point-by-point response to your comments:

1. Yes, you are right that the slope unit is m/m. We added this unit in the Abstract and maintain its absolute values rather than using degrees. This is because it is almost a consensus that in a very low slope environment like river deltas, many publications have used absolute values or order of magnitudes of $10^{-3}$-$10^{-5}$ rather than using degrees as in alluvial environments. Having the absolute values will also put more details in choosing the slope values we used in our models.
2. Rephrased as suggested on L108.
3. Rewritten as suggested on L404.

We would like to thank you for your time and effort spent on our work.

Best wishes,
Octria Adi Prasojo

5 March 2025